# Improving Causal Inference Robustness via Reinforcement-guided Diffusion Models

## Abstract

Estimating the Conditional Average Treatment Effect (CATE) is essential to personalized decision-making in causal inference. However, in real-world practices, CATE models often suffer degraded performance when faced with unknown distribution shifts between training and deployment environments. To tackle this challenge, we introduce **C**ausal **A**dversarial **R**einforcement-guided **D**iffusion (**CARD**), a model-agnostic framework that can be wrapped around any existing CATE learner to improve its robustness against unknown distribution shifts. CARD formulates the CATE modeling process as a minimax game: a reinforcement learning agent guides a diffusion model to generate adversarial data augmentations that maximize the CATE learner's loss, and then the learner is trained to minimize this worst-case loss, creating a principled robust optimization procedure. The comprehensive experimental results demonstrate that CARD consistently improves the robustness of diverse CATE learners against challenging data corruptions, including measurement error, missing values, and unmeasured confounding, confirming its broad applicability and effectiveness.

## 1 Introduction

Estimating the Conditional Average Treatment Effect (CATE) is a core problem in causal inference, as it quantifies how an intervention would differentially affect subgroup-level (or approximated individual-level) as a function of observed covariates, enabling personalized decision-making in various domains such as (Farrell, 2015; Chernozhukov et al., 2018; Kitagawa & Tetenov, 2018; Abadie et al., 2023), statistics (Wager & Athey, 2018; Li & Wager, 2022; Foster & Syrgkanis, 2023; Kennedy, 2023), clinical (Zhang et al., 2019; Qian et al., 2021; Bica et al., 2021; Kinyanjui & Johansson, 2022; Feuerriegel et al., 2024; Ma et al., 2025), and financial application (Li et al., 2023; Huang et al., 2023b; Fernández-Loría et al., 2023; Wu et al., 2025a). The practical of CATE models often hinges on a crucial but fragile assumption: *external validity (or transportability)* (Pearl & Bareinboim, 2011; Bareinboim & Pearl, 2016). That is, causal conclusions derived from a source environment must remain valid when the model is deployed in a different target population.

However, in real-world practice, this external validity assumption is often violated. CATE models can suffer a degraded performance when confronted when unknown *distribution shifts* between the training and deployment environments present, which is often incurred by data imperfections (Kallus et al., 2018; Agarwal & Singh, 2021; Zhang et al., 2023), such as measurement error (Imai & Yamamoto, 2010; Battistin & Chesher, 2014; Kuroki & Pearl, 2014; Pei et al., 2019), missing values (Rubin, 1976; Bang & Robins, 2005; Mohan et al., 2013; Yang et al., 2019; Mayer et al., 2020), and unmeasured confounding (Kallus et al., 2019; Ding et al., 2022; Oprescu et al., 2023; Xiao et al., 2024; Dorn et al., 2025). Such data imperfections signify structural discrepancies between the source and target domains, thereby rendering the direct transfer of causal conclusions invalid. We demonstrate this challenge with the following motivating example.

**Motivating example.** Suppose a technical company trains a CATE model on its proprietary clean and well structured dataset, and intend to license it to a hospital system. The model must be validated on the hospital's own Electronic Health Record (EHR) data, which constitutes an unseen target domain as it was unavailable during the initial training phase. The EHR data reflects a different data generating process with systematic imperfections that induce a distribution shift from

the source data (Ruan et al., 2024): (i) measurement error, arising from device inaccuracies, reporting biases, or procedural variability; (ii) missing values, caused by privacy restrictions, legal constraints, or transcription errors; and (iii) unmeasured confounders, such as socioeconomic status, environmental exposures, or genetic predispositions, can trigger severe concept drift. These uncertain data imperfections significantly hinder the generalization of the CATE model trained in the source domain to the EHR data.

To tackle the distribution shift problem in causal inference, existing methods develop specialized causal estimators for specific data shift types. Examples include CATE learners that aim to be robust to covariate shift (Kern et al., 2024) or to concept drift (Zhang et al., 2024), and policy learning methods that seek robustness under combined and unknown shifts (Kallus et al., 2022; Mu et al., 2022; Si et al., 2023). Most of these studies are grounded in structural assumptions about the shift or the estimator itself, and their advancements in robust causal learning motivates a complementary question: *Can we develop a method that is capable to enhance the robustness of any existing CATE learner to unknown distribution shifts without requiring additional structural assumptions?*

Inspired by this question, we propose **C**ausal **A**dversarial **R**einforcement-guided **D**iffusion (**CARD**), a model-agnostic framework that strengthens the robustness of *any* existing CATE learner without redesigning its internal architecture. CARD frames robust CATE training as a minimax game between an adversarial generator and a CATE learner. In CARD, specifically, a reinforcement learning agent guides a diffusion model to produce adversarial proxies that maximally challenge the CATE learner, and the learner then adapts by minimizing the worst-case error over these generated augmentations, yielding a principled robust optimization routine tailored to CATE estimation.

Our main **contributions** are threefold:

- We propose a novel model-agnostic framework, CARD, which can be flexibly integrated with any existing CATE learner to enhance its robustness against a wide range of unknown distribution shifts, without requiring additional structural assumptions or prior knowledge of target information.
- To the best of our knowledge, we are the first to introduce a reinforcement-learning guided diffusion model in causal inference literature. This might bring new possibilities for other causal inference tasks, such as counterfactual generation (Yoon et al., 2018), dimension reduction (Liu et al., 2024), and model evaluation (Athey et al., 2024), among others.
- We empirically demonstrate that CARD consistently improves the robustness of popular CATE learners when deployed in challenging target data corruptions, involving measurement error, missing values, and unmeasured confounding, confirming its reliability and adaptability to real-world causal inference tasks.

## 2 RELATED WORK

**Data combination and external validity.** A central challenge in causal inference is to generalize effects learned in the source dataset to a target population. This problem is formalized under external validity (or transportability) (Pearl & Bareinboim, 2011; Bareinboim & Pearl, 2016). Data combination frameworks specify when and how evidence from multiple sources can be fused across populations to identify causal quantities, explicitizing the role of distributional differences across domains (Bareinboim & Pearl, 2016; Dahabreh & Hernán, 2019). When the transportability assumption holds, combining data can improve the precision and efficiency of treatment effect estimation (Hatt et al., 2022; Dahabreh et al., 2023; Huang et al., 2023a; Wu et al., 2025b; Rudolph et al., 2025). In practice, however, the transportability assumption is often violated due to unobserved heterogeneity between the source and target domains. To address this, statistical work develops sensitivity analysis and partial identification tools for average treatment effects (Nie et al., 2021; Huang, 2024; Yadlowsky et al., 2022). On the modeling side, there is growing interest in conditional effect estimation without assuming transportability. Several studies address latent confounding by combining RCT and observational data, proposing methods such as the integrative R-learner (Wu & Yang, 2022) and the MetaDebias neural network (Xiao et al., 2024).

**Causal inference under distribution shift.** A growing body of research has examined how to make causal inference methods robust when the deployment distribution differs from the training

environment. Existing approaches can be broadly categorized into two lines. The first line focuses on robust CATE estimation under specific types of shifts. For example, recent work addresses co-variate shift by controlling worst-case bias across target covariate distributions (Jeong & Namkoong, 2020) or by enforcing multi-accuracy constraints on CATE learners (Kern et al., 2024). Other stud-ies primarily focus on concept drift, for instance, by optimizing the CATE function within an un-certainty set over convex combinations of multisite CATE functions under a known target covariate distribution (Zhang et al., 2024). The second line of work emphasizes causal decision making via robust optimization, which aims to learn treatment assignment rules that remain effective to unseen confounding scenarios or target environments (Kallus & Zhou, 2021; Kallus et al., 2022; Mu et al., 2022; Kido, 2022; Si et al., 2023; Shen et al.; Wang et al.; Hess et al., 2025). While powerful for de-riving robust policies, these approaches are primarily designed for policy learning rather than CATE estimation. This gap underscores the necessity of developing methods specifically tailored for gen-eralizing CATE estimation to unseen target domains without requiring prior knowledge of covariate distributions or treatment effect heterogeneity.

## 3 PROBLEM SETUP

This study is grounded in the potential outcome framework (Rubin, 1974; 2005). Let $\{(X_i, A_i, Y_i)\}_{i=1}^n$ denote an observational sample of $n$ i.i.d. units drawn from a *source* popula-tion. For unit $i$, $X_i \in \mathcal{X} \subset \mathbb{R}^d$ is a $d$-dimensional pre-treatment covariate vector, $A_i \in \{0, 1\}$ is a binary treatment indicator, and $\{Y_i^0, Y_i^1\}$ are the corresponding potential outcomes. The observed (factual) outcome is $Y_i = Y_i^{A_i}$, and the unobserved (counterfactual) outcome is $Y_i^{1-A_i}$. Our target estimand is the CATE, which captures the sub-population treatment heterogeneity:

$$\tau(x) := \mathbb{E}\big[ Y^1 - Y^0 \,\big|\, X = x \big]. \tag{1}$$

Estimating $\tau(x)$ from observational data presents a key challenge, due to the fundamental problem of causal inference: for any unit, only one potential outcome can be observed. To identify the CATE from observational data in the source domain, we rely on the following standard assumptions.

**Assumption 1** (SUTVA, Consistency, and Overlap). *For all units in the source domain, we have the following assumptions: **Consistency & SUTVA:** The observed outcome for unit $i$ receiving treatment $a$ is the potential outcome $Y^a$, and potential outcomes of this unit are not affected by the treatment assignments of other units. **Overlap (Positivity):** The probability of receiving treatment is bounded away from 0 and 1 for all covariate profiles, i.e., $0 < P(A = 1|X = x) < 1$ for all $x \in \mathcal{X}$. **Internal validity (Unconfoundedness):** The treatment assignment is independent of the potential outcomes, conditional on the observed covariates, i.e., $\{Y^1, Y^0\} \perp\!\!\!\perp A \mid X$.*

### 3.1 GENERALIZING CATE UNDER DISTRIBUTION SHIFT

A critical generalization challenge arises when an estimator $\hat{\tau}(x)$, trained on the source domain ($P_\mathcal{S}$), must be deployed in an unseen target domain ($P_\mathcal{T}$). The CATE model's performance in this new domain is threatened by potential distribution shifts in two main types.

**Covariate shift.** The most common and well-studied type of distribution shift is covariate shift, where the marginal distribution of covariates differs across domains, i.e., $P_\mathcal{S}(X) \neq P_\mathcal{T}(X)$. Gen-eralization under this shift is made possible by the transportability assumption.

**Assumption 2** (External validity (Transportability)). *The conditional distribution of potential out-comes given covariates is invariant across domains, i.e., $P_\mathcal{S}(Y^a|X) = P_\mathcal{T}(Y^a|X)$ for $a \in \{0, 1\}$.*

This assumption implies that the underlying causal mechanisms are stable across domains, and thus the true CATE function is the same in both domains: $\tau_\mathcal{S}(x) = \tau_\mathcal{T}(x)$. Nevertheless, even with transportability, CATE model performance can deteriorate when deploying $\hat{\tau}$ in the target domain due to the covariate distribution mismatch, which is a common issue in machine learning studies.

**Concept drift.** A more severe challenge arises from concept drift, where the transportability as-sumption is violated, meaning $P_\mathcal{S}(Y^a|X) \neq P_\mathcal{T}(Y^a|X)$. As highlighted in our motivating exam-ple, such drift is often caused by unmeasured confounders present only in the target domain, which alter the treatment heterogeneity. Under concept drift, the true CATE function is no longer invariant across domains, i.e., $\tau_\mathcal{S}(x) \neq \tau_\mathcal{T}(x)$, making out-of-domain CATE estimation substantially more challenging than the covariate-shift-only setting.

## 4 METHOD

In this section, we introduce our proposed framework, **C**ausal **A**dversarial **R**einforcement-guided **D**iffusion (**CARD**). We begin by formulating robust CATE estimation as a minimax optimization problem. We then detail how a reinforcement learning agent guides a diffusion model to generate adversarial proxies that realize this objective. Finally, we present the detailed training pipeline for integrating CARD with any CATE estimator.

### 4.1 A MINIMAX OBJECTIVE FOR ROBUST CATE ESTIMATION

Given source data samples $(X, A, Y) \sim P_{\mathcal{S}}$, a standard CATE learner $f_\phi$ with parameters $\phi$ is trained by minimizing an objective $\mathcal{L}^{\text{inf}}(\phi) = \mathbb{E}[\ell(X, A, Y; f_\phi)]$, where $\ell$ is the loss function associated with the chosen meta-learner. To protect the estimator against unknown distribution shifts in the target domain, we optimize $f_\phi$ with a new objective $\mathcal{L}^{\text{inf}}(\phi, Z)$ in a robust optimization manner:

$$\min_\phi \max_{Z \in \Omega} \mathcal{L}^{\text{inf}}(\phi, Z) := \mathbb{E}_{(X,A,Y) \sim P_{\mathcal{S}}} \big[ \ell(X \oplus Z, A, Y; f_\phi) \big], \tag{2}$$

where $\Omega$ is an uncertainty set defining the space of learnable adversarial proxies. Conceptually, solving this objective forces the inferencer $f_\phi$ to be robust against the most harmful proxies in $\Omega$. While some causal inference studies formulate similar adversarial problems as distributionally robust optimization (DRO), they often define $\Omega$ based on statistical distances or strong structural assumptions, as discussed in Section 2. Our key departure is that the inferencer $f_\phi$ is trained on covariates augmented by learned adversarial proxies, creating a more flexible robustness mechanism without loss of original covariate information.

### 4.2 ROBUST CATE ESTIMATION WITH CARD

Instead of constraining the adversary to a predefined uncertainty set $\Omega$ (e.g., a KL-ball (Kallus et al., 2022; Si et al., 2023)), we design a framework that learns to generate worst-case proxies $Z$ using a score-based diffusion model guided by a reinforcement learning (RL) agent.

**Score-based diffusion.** A score-based diffusion model (Song et al.) consists of a forward process that progressively adds noise to data $Z_0$ over a time interval $t \in [0, T]$, governed by a stochastic differential equation (SDE):

$$dZ = f(Z, t)dt + g(t)dW_t, \tag{3}$$

where $f(Z, t)$ is the drift coefficient, $g(t)$ is the diffusion coefficient, and $W_t$ is a standard Wiener process. The corresponding reverse process generates data by traversing time from $T$ to $0$. Using the Fokker-Planck equation of the marginal density (Suh et al.), the reverse-time SDE is:

$$dZ = \big[ f(Z, t) - g(t)^2 \, \nabla_z \log p_t(Z) \big] dt + g(t) \, d\bar{W}_t, \tag{4}$$

where $\bar{W}_t$ is a Wiener process running backward from $t = T$ to $t = 0$. The score $\nabla_z \log p_t(Z)$ is approximated by a neural network $g_\theta(z, t)$, pretrained with the standard score-matching objective:

$$\mathcal{L}^{\text{diff}}(\theta) = \mathbb{E}_{Z_0, Z_t \sim p_t(\cdot|Z_0), t \sim \mathcal{U}[\varepsilon, T]} \big[ \lambda(t)^2 \| g_\theta(Z_t, t) - \nabla_z \log p_t(Z_t \mid Z_0) \|_2^2 \big], \tag{5}$$

where $\lambda(t) > 0$ weights time steps and $\varepsilon > 0$ ensures numerical stability. In practice, the diffusion model is often applied on a low-dimensional latent code $Z$ obtained from an autoencoder.

**Reinforcement-guided adversarial generation.** To solve the inner maximization of Eqn. (2), we frame the reverse diffusion process as a Markov Decision Process (MDP) (Black et al.) and use an RL agent to steer the generation toward adversarial proxies. The objective is to guide the score model $g_\theta$ to maximize the expected cumulative reward along the denoising trajectory:

$$\mathcal{J}(\theta) = \mathbb{E} \Big[ \sum_{t=1}^T \log g_\theta(Z_{t-1}|Z_t) \frac{G_t - \mu_G}{\sigma_G} \Big], \quad \text{where} \quad G_t = \sum_{k=t}^T \gamma^{k-t} \mathcal{L}^{\text{inf}}(\phi, Z). \tag{6}$$

Here, $\mathcal{L}^{\text{inf}}(\phi, Z)$ is the immediate reward at step $t$, $G_t$ is the discounted return from step $t$ with discount factor $\gamma \in (0, 1]$, and the returns are standardized per-trajectory with the mean $\mu_G$ and

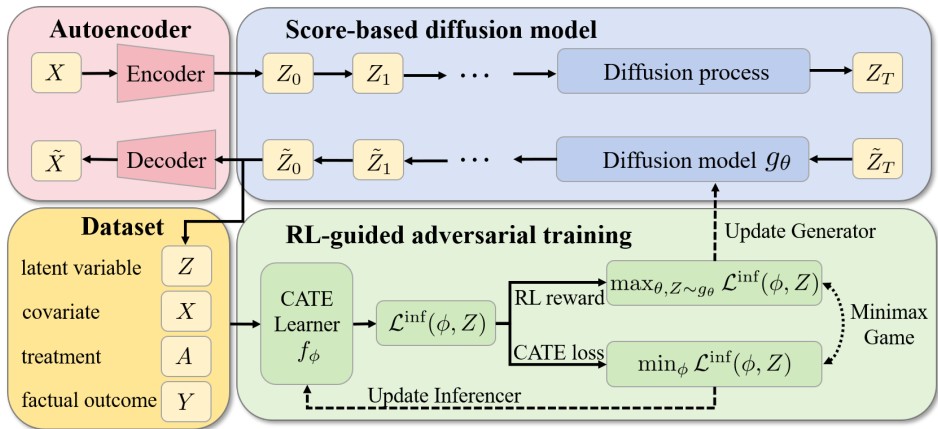

Figure 1: An overview of the proposed CARD training pipeline for robust CATE estimation.

standard deviation $\sigma_G$. To maintain generation quality and stabilize training, we combine this RL objective with the original score-matching loss, yielding the final objective for the generator:

$$\max_{\theta} \mathcal{L}^{\text{full}}(\theta) = \mathcal{J}(\theta) - \alpha \mathcal{L}^{\text{diff}}(\theta), \tag{7}$$

where $\alpha > 0$ is a balancing hyperparameter. This fine-tuning process transforms the diffusion model from a simple data generator into a sophisticated adversary capable of generating worst-case proxies that approximate the solution to the inner maximization in our minimax objective (2).

**Algorithm of CATE learning with CARD.** The complete procedure of training CATE with CARD, which alternates between updating the generator and the inferencer, is outlined in Algorithm 1. We also inllustrate the corresponding pipeline in Figure 1.

---

**Algorithm 1** CATE model training with CARD
***
**Require:** Source data $(X, A, Y) \sim P_{\mathcal{S}}$, inferencer (base CATE learner) $f_\phi$, Autoencoder $(\text{Enc}_\psi, \text{Dec}_\psi)$, diffusion model $g_\theta$.
1: **Phase 1: Pre-training**
2: Train autoencoder on covariates $X$ to learn a latent space $Z$.
3: Pre-train diffusion model $g_\theta$ on latent representations $Z = \text{Enc}_\psi(X)$ via Eqn. (5).
4: **Phase 2: CATE model training with minimax**
5: **for** each training epoch $e = 1, \ldots, E$ **do**
6:    Sample $Z_T \sim \mathcal{N}(0, I)$ and generate a denoising trajectory $(Z_T, \ldots, Z_0)$ using $g_\theta$.
7:    For each sample, compute trajectory returns $G_t$ via Eqn. (6).
8:    Update generator parameters $\theta$ by maximizing $\mathcal{L}^{\text{full}}(\theta)$ from Eqn. (7).
9:    Update inferencer parameters $\phi$ by minimizing $\mathcal{L}^{\text{inf}}(\phi, Z_0)$ from Eqn. (2).
**Ensure:** Generator $g_\theta$ for $Z_0$ generation, and inferencer $f_\phi(X \oplus Z_0, A, Y)$ for CATE estimation.

---

## 5 EXPERIMENTS

### 5.1 EXPERIMENTAL SETUP

**CATE learners.** Our evaluation includes eight prominent CATE estimation methods, comprising five meta-learners and three specialized neural network models. The meta-learners represent a diverse set of strategies, including indirect-type (S-learner and T-Learner) and direct-type (X-learner (Künzel et al., 2019), DR-learner (Kennedy, 2023; Foster & Syrgkanis, 2023), and R-Learners (Nie & Wager, 2021)) approaches. The representation-based models consist of widely-recognized architectures: TARNet & CFR-Wass (Shalit et al., 2017; Johansson et al., 2022) and DragonNet (Shi et al., 2019). The specific details of implementing these CATE learners with CARD are presented in Section A.1, and detailed parameter configurations for all models are provided in Appendix A.2.

**Experimental settings.** Evaluating CATE estimators requires access to ground-truth treatment effects, which are unavailable in real-world data. Therefore, following established practice in causal inference research (Curth & Van der Schaar, 2021; Curth & Van Der Schaar, 2023; Huang et al., 2024), we employ a semi-synthetic data generating process with covariates collected from ACIC2016 dataset (Dorie et al., 2019). The dataset contains 4802 samples with $d = 22$ continuous covariates. The treatment assignment $A_i$ is generated from a Bernoulli distribution based on the covariates $A_i|X_i \sim \text{Bern}\left(1/(1 + \exp(-(\beta'_T X_i)))\right)$. The potential outcome generation is based on a additive interaction terms, with a complex quadratic heterogeneous treatment effects:

$$Y_i = \sum_{j}^{d} \beta'_j X_{i;j} + \sum_{j=1}^{d} \sum_{k=j}^{d} \beta'_{j,k} X_{i;j} X_{i;k} + A_i \sum_{j=1}^{d} \sum_{k=j}^{d} \gamma_{i,j} X_{i;j} X_{i;k} + \epsilon_i. \tag{8}$$

The coefficients are set as: $\beta'_T \sim Bern(0.1)$, $\beta'_j \sim Bern(0.5)$, $\beta'_{j,k} \sim Bern(0.5)$, $\gamma_{i,j} \sim Bern(0.1)$, and the noise term $\epsilon_i$ is sampled from $\mathcal{N}(0, 0.1)$. We repeat the above data generating process to generate 30 distinct datasets, each partitioned into training/validation/testing ratio of 49%/21%/30%.

**Distribution shift settings.** To evaluate the robustness of CARD, we introduce three types of distribution shifts exclusively in the test set:

- **Measurement error**: We simulate measurement error by adding Gaussian noise to the covariates of the test set, while the underlying data generating process remains unchanged. The observed covariates become $X_i^{obs} = X_i + \mathcal{N}(0, \delta^2 I_d)$, where $\delta$ controls the shift level incurred by measurement error and varies across $\{0.1, 0.5, 1.0, 1.5, 2.0, 2.5, 3.0\}$.

- **Missing values**: This setting introduces missingness to the covariates of the test set, while the the underlying data generating process remains unchanged. We apply a binary mask to the covariates, where each element is independently set to $0$ with probability $\rho$. We use the MICE algorithm (Kallus et al., 2018) to enable CATE to be deployed on incomplete test data. The missingness rate $\rho$ controls the shift level incurred by missing values and varies across $\{0.01, 0.05, 0.1, 0.2, 0.3, 0.4, 0.5\}$.

- **Unmeasured confounding**: This scenario introduces a shift in the outcome generation mechanism for the test set. The observed covariates $X_{obs}$ remain unchanged, but the potential outcomes are generated by a new model that includes both observed confounders $X_{obs}$ and hidden confounders $U$, i.e., $X = (X_{obs}, U)$ in Eqn. (8). The unmeasured confounders are drawn from uniform distribution $U_{i,j} \sim \mathcal{U}(-3, 3)$, and the dimension of hidden confounders $d^U$ is varied across $\{1, 5, 10, 15, 20, 25, 30\}$.

**Evaluation criteria.** We evaluate model performances using the Precision in Estimation of Heterogeneous Effect (PEHE) (Hill, 2011), a standard metric that measures the root mean squared error between the estimated and true CATE values, denoted by $\epsilon_{\text{PEHE}}(\hat{\tau})$. And we use $\epsilon_{\text{RI}}(\hat{\tau})$ to denote the relative improvement of a base CATE learner $\hat{\tau}$ when it is trained with CARD, i.e., the CATE learner with CARD $\hat{\tau}^{\text{CARD}}$, denoted by $\epsilon_{\text{RI}}(\hat{\tau})$.

$$\epsilon_{\text{PEHE}}(\tau) = \sqrt{\frac{1}{n} \sum_{i=1}^{n} \left(\tau(X_i) - \tau_{\text{true}}(X_i)\right)^2},$$

where $\tau$ and $\tau_{\text{true}}$ are arbitrary CATE model and the ground-truth CATE function, respectively. To specifically quantify the benefit of our method, we also introduce the Relative Improvement (RI) in PEHE. This metric calculates the percentage reduction in PEHE achieved by applying our CARD framework to a base CATE estimator:

$$\epsilon_{\text{RI}}(\hat{\tau}) = \frac{\epsilon_{\text{PEHE}}(\hat{\tau}) - \epsilon_{\text{PEHE}}(\hat{\tau}^{\text{CARD}})}{\epsilon_{\text{PEHE}}(\hat{\tau})},$$

where $\epsilon_{\text{PEHE}}(\hat{\tau})$ is the PEHE of the original CATE learner and $\epsilon_{\text{PEHE}}(\hat{\tau}^{\text{CARD}})$ is the PEHE of the same estimator after being trained with CARD.

Table 1: Comparison of **average PEHE** over 30 runs for various CATE learners, with and without the CARD framework, under three distribution shift scenarios: measurement error, missing values, and unmeasured confounding. Bold denotes the better results for each learner pair.

| Settings | Measurement error (controlled by $\delta$) | | | | | | | Missing values (controlled by $\rho$) | | | | | | | Unmeasured confounding (controlled by $d^U$) | | | | | | |
|---|---|---|---|---|---|---|---|---|---|---|---|---|---|---|---|---|---|---|---|---|---|
| Bias level | 0.1 | 0.5 | 1.0 | 1.5 | 2.0 | 2.5 | 3.0 | 0.01 | 0.05 | 0.1 | 0.2 | 0.3 | 0.4 | 0.5 | 1 | 5 | 10 | 15 | 20 | 25 | 30 |
| S-learner | 0.578 | 0.632 | 0.761 | 0.889 | 0.981 | 1.041 | 1.080 | 0.577 | 0.581 | 0.586 | 0.598 | 0.622 | 0.649 | 0.671 | 0.662 | 0.956 | 1.562 | **1.883** | **2.139** | 2.453 | **2.862** |
| S+CARD | **0.565** | **0.612** | **0.722** | **0.838** | **0.927** | **0.988** | **1.028** | **0.564** | **0.569** | **0.574** | **0.589** | **0.611** | **0.636** | **0.661** | **0.644** | **0.939** | **1.560** | 1.884 | 2.141 | **2.452** | 2.863 |
| T-learner | 1.105 | 1.283 | 1.833 | 2.669 | 3.653 | 4.705 | 5.792 | 1.096 | 1.085 | 1.072 | 1.056 | 1.065 | 1.057 | 1.014 | 1.240 | 1.420 | 1.902 | 2.187 | 2.396 | 2.693 | 3.069 |
| T+CARD | **1.043** | **1.201** | **1.687** | **2.416** | **3.290** | **4.243** | **5.239** | **1.039** | **1.031** | **1.005** | **0.994** | **1.006** | **1.001** | **0.967** | **1.207** | **1.381** | **1.873** | **2.133** | **2.355** | **2.651** | **3.034** |
| X-learner | 0.669 | 0.734 | 0.921 | 1.209 | 1.568 | 1.973 | 2.408 | 0.667 | 0.669 | 0.672 | 0.683 | 0.702 | 0.723 | 0.731 | 0.771 | 1.033 | 1.647 | 1.955 | 2.193 | 2.503 | 2.902 |
| X+CARD | **0.610** | **0.670** | **0.844** | **1.113** | **1.453** | **1.848** | **2.277** | **0.608** | **0.609** | **0.615** | **0.629** | **0.654** | **0.681** | **0.695** | **0.738** | **1.010** | **1.627** | **1.936** | **2.180** | **2.490** | **2.891** |
| R-learner | 0.825 | 0.889 | 1.082 | 1.379 | 1.741 | 2.143 | 2.567 | 0.822 | 0.819 | 0.815 | 0.813 | 0.818 | 0.820 | 0.818 | 0.847 | 1.053 | 1.746 | 2.051 | 2.267 | 2.549 | 2.962 |
| R+CARD | **0.760** | **0.818** | **0.983** | **1.234** | **1.542** | **1.887** | **2.258** | **0.759** | **0.761** | **0.761** | **0.763** | **0.775** | **0.785** | **0.794** | **0.820** | **1.027** | **1.722** | **2.025** | **2.242** | **2.538** | **2.949** |
| DR-learner | 0.755 | 0.861 | 1.169 | 1.632 | 2.193 | 2.810 | 3.460 | 0.751 | 0.747 | 0.746 | 0.752 | 0.766 | 0.782 | 0.783 | 0.906 | 1.142 | 1.701 | 2.001 | 2.237 | 2.546 | 2.939 |
| DR+CARD | **0.693** | **0.797** | **1.095** | **1.544** | **2.094** | **2.701** | **3.343** | **0.690** | **0.687** | **0.687** | **0.697** | **0.721** | **0.741** | **0.747** | **0.872** | **1.123** | **1.675** | **1.971** | **2.217** | **2.530** | **2.920** |
| TARNet | 0.675 | 0.776 | 1.080 | 1.559 | 2.164 | 2.843 | 3.564 | 0.670 | 0.670 | 0.671 | 0.683 | 0.708 | 0.731 | 0.731 | 0.842 | 1.109 | 1.612 | 1.903 | 2.173 | 2.499 | 2.881 |
| TAR+CARD | **0.643** | **0.728** | **0.984** | **1.376** | **1.879** | **2.452** | **3.074** | **0.639** | **0.639** | **0.642** | **0.657** | **0.684** | **0.710** | **0.717** | **0.811** | **1.077** | **1.598** | **1.896** | **2.163** | **2.486** | **2.875** |
| CFR-Wass | 0.707 | 0.765 | 0.965 | 1.336 | 1.840 | 2.409 | 3.009 | 0.693 | 0.694 | 0.705 | 0.715 | 0.728 | 0.746 | 0.744 | 0.771 | 1.040 | 1.647 | 1.936 | 2.187 | 2.498 | 2.898 |
| CFR+CARD | **0.675** | **0.736** | **0.927** | **1.258** | **1.722** | **2.270** | **2.856** | **0.666** | **0.664** | **0.671** | **0.684** | **0.707** | **0.727** | **0.733** | **0.767** | **1.036** | **1.620** | **1.934** | **2.176** | **2.493** | **2.894** |
| DragonNet | 0.689 | 0.780 | 1.058 | 1.502 | 2.059 | 2.682 | 3.343 | 0.686 | 0.685 | 0.683 | 0.695 | 0.716 | 0.736 | 0.738 | 0.832 | 1.084 | 1.640 | 1.945 | 2.193 | 2.512 | 2.910 |
| Dragon+CARD | **0.659** | **0.747** | **1.014** | **1.429** | **1.950** | **2.533** | **3.156** | **0.656** | **0.657** | **0.662** | **0.674** | **0.700** | **0.723** | **0.725** | **0.827** | **1.083** | **1.633** | **1.934** | **2.186** | **2.498** | **2.899** |

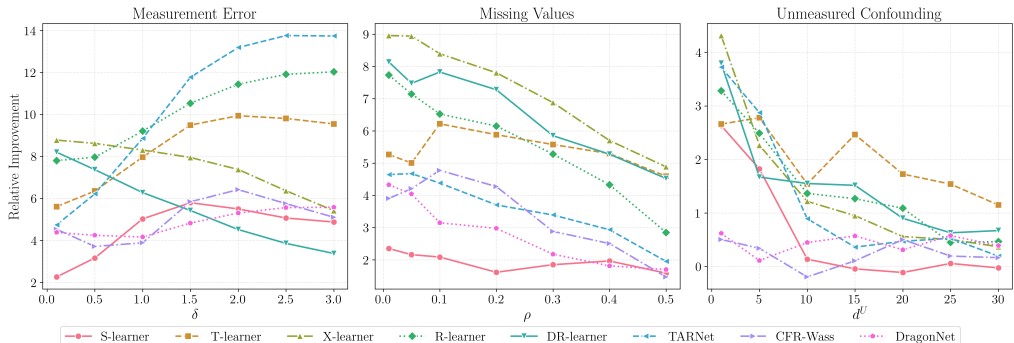

Figure 2: Performance gains from CARD: Relative Improvement in **average PEHE** over 30 runs for various CATE learners under three distribution shift scenarios. The value is in percentage.

## 5.2 EXPERIMENTAL RESULTS

In this section, we evaluate the efficacy of CARD by investigating two primary research questions: (1) whether CARD improves the average performance of CATE learners under distribution shifts, and (2) whether it enhances their worst-case performance under distribution shifts.

### 5.2.1 AVERAGE PERFORMANCE ANALYSIS

The experimental results for average PEHE are presented in Table 1, with the Relative Improvement (RI) from CARD visualized in Figure 2. Our analysis reveals several key insights into the CARD framework's effectiveness.

**CARD consistently enhances performance across diverse learners and shifts.** The results demonstrate a near-universal improvement in average PEHE when CARD is applied. Across all eight CATE learners and three distinct types of data corruption, the "+CARD" variants consistently outperform their standard counterparts. This broad applicability holds for both traditional meta-learners (e.g., S-learner, T-learner) and more complex representation-based models (e.g., TARNet, DragonNet). The framework's benefits extend to models with varying degrees of baseline robustness. For instance, under severe measurement error ($\delta = 3.0$), CARD reduces the PEHE of the highly vulnerable T-learner from 5.792 to 5.239 (a 9.5% relative improvement). Simultaneously, it also enhances the relatively robust S-learner, reducing its PEHE from 1.080 to 1.028, highlighting the general effectiveness of CARD.

**The magnitude of CARD's improvement is context-dependent.** CARD yields benefits across all evaluated settings, and more specifically, the magnitude of improvement depends on the type and severity of the distribution shift. As shown in Figure 2, CARD produces its largest and most

Table 2: Comparison of **worst-case PEHE** over 30 runs for various CATE learners, with and without the CARD framework, under three distribution shift scenarios: measurement error, missing values, and unmeasured confounding. Bold denotes the better results for each learner pair.

| Settings | Measurement error (controlled by $\delta$) | | | | | | | Missing values (controlled by $\rho$) | | | | | | | Unmeasured confounding (controlled by $d^U$) | | | | | | |
|---|---|---|---|---|---|---|---|---|---|---|---|---|---|---|---|---|---|---|---|---|---|
| Bias level | 0.1 | 0.5 | 1.0 | 1.5 | 2.0 | 2.5 | 3.0 | 0.01 | 0.05 | 0.1 | 0.2 | 0.3 | 0.4 | 0.5 | 1 | 5 | 10 | 15 | 20 | 25 | 30 |
| S-learner | 1.020 | 1.058 | 1.173 | 1.448 | 1.746 | 1.961 | 2.118 | 1.023 | 1.024 | 1.032 | 1.043 | 1.071 | 1.105 | 1.128 | 0.942 | 1.282 | 1.897 | **2.083** | **2.296** | 2.525 | **3.010** |
| S+CARD | **1.024** | **1.040** | **1.108** | **1.157** | **1.401** | **1.581** | **1.706** | **1.026** | **1.029** | **1.037** | **1.050** | **1.067** | **1.090** | **1.119** | **0.826** | **1.123** | **1.880** | 2.004 | 2.259 | **2.498** | 2.985 |
| T-learner | 2.507 | 2.781 | 3.617 | 4.953 | 6.574 | 8.315 | 10.109 | 2.483 | 2.470 | 2.395 | 2.335 | 2.406 | 2.250 | 2.064 | 2.473 | 2.541 | 2.948 | 3.122 | 3.269 | 3.585 | 3.792 |
| T+CARD | **1.453** | **1.631** | **2.252** | **3.207** | **4.386** | **5.659** | **7.008** | **1.463** | **1.451** | **1.413** | **1.404** | **1.440** | **1.366** | **1.374** | **1.558** | **1.624** | **2.037** | **2.279** | **2.550** | **2.906** | **3.203** |
| X-learner | 1.447 | 1.478 | 1.688 | 2.096 | 2.737 | 3.707 | 4.729 | 1.441 | 1.445 | 1.381 | 1.396 | 1.400 | 1.373 | 1.306 | 1.247 | 1.424 | 2.283 | 2.285 | 2.489 | 2.762 | 3.070 |
| X+CARD | **1.294** | **1.310** | **1.260** | **1.622** | **1.344** | **1.722** | **2.105** | **1.288** | **1.295** | **1.245** | **1.258** | **1.264** | **1.239** | **1.200** | **0.973** | **1.177** | **2.165** | 2.382 | 2.595 | 2.632 | 3.021 |
| R-learner | 1.389 | 1.473 | 2.053 | 2.986 | 4.110 | 5.327 | 6.598 | 1.386 | 1.385 | 1.354 | 1.375 | 1.395 | 1.388 | 1.351 | 1.368 | 1.398 | 2.309 | 2.613 | 2.719 | 2.829 | 3.331 |
| R+CARD | **1.190** | **1.563** | **1.818** | **2.400** | **3.177** | **4.048** | **5.008** | **1.187** | **1.207** | **1.164** | **1.151** | **1.166** | **1.157** | **1.137** | **1.444** | **1.628** | **2.406** | **2.580** | **2.764** | **2.909** | **3.245** |
| DR-learner | 1.598 | 1.677 | 2.097 | 2.723 | 3.463 | 4.809 | 6.251 | 1.592 | 1.594 | 1.500 | 1.529 | 1.533 | 1.495 | 1.398 | 1.498 | 1.645 | 2.410 | 2.298 | 2.534 | 2.899 | 3.137 |
| DR+CARD | **1.485** | **1.588** | **1.966** | **2.517** | **3.240** | **4.160** | **5.462** | **1.479** | **1.481** | **1.360** | **1.369** | **1.392** | **1.351** | **1.314** | **1.382** | **1.548** | **2.330** | **2.223** | **2.463** | **2.831** | **3.060** |
| TARNet | 1.214 | 1.234 | 1.607 | 2.837 | 4.566 | 6.568 | 8.715 | 1.213 | 1.213 | 1.172 | 1.174 | 1.177 | 1.176 | 1.156 | 1.394 | 1.621 | 2.112 | 2.193 | 2.372 | 2.698 | 3.122 |
| TAR+CARD | **1.192** | **1.224** | **1.034** | **1.623** | **2.479** | **3.516** | **4.686** | **1.193** | **1.195** | **1.175** | **1.178** | **1.186** | **1.178** | **1.163** | **0.943** | **1.244** | **2.030** | **2.032** | **2.343** | **2.540** | **3.015** |
| CFR-Wass | 1.495 | 1.536 | 1.656 | 2.956 | 4.953 | 7.314 | 9.834 | 1.489 | 1.487 | 1.391 | 1.398 | 1.384 | 1.364 | 1.339 | 1.502 | 1.724 | 2.336 | 2.180 | 2.423 | 2.658 | 3.095 |
| CFR+CARD | **1.401** | **1.436** | **1.535** | **1.738** | **2.814** | **4.131** | **5.565** | **1.399** | **1.397** | **1.319** | **1.335** | **1.324** | **1.309** | **1.265** | **0.976** | **1.288** | **2.175** | **1.940** | **2.261** | **2.446** | **2.927** |
| DragonNet | 1.508 | 1.539 | 1.659 | 2.481 | 3.547 | 4.867 | 6.447 | 1.505 | 1.503 | 1.396 | 1.412 | 1.415 | 1.388 | 1.320 | 1.265 | 1.634 | 2.326 | 2.277 | 2.424 | 2.738 | 3.100 |
| Dragon+CARD | **1.269** | **1.295** | **1.361** | **2.422** | **3.376** | **5.429** | **7.072** | **1.262** | **1.265** | **1.220** | **1.221** | **1.225** | **1.205** | **1.172** | **1.526** | **1.740** | **2.145** | **2.011** | **2.238** | **2.589** | **3.059** |

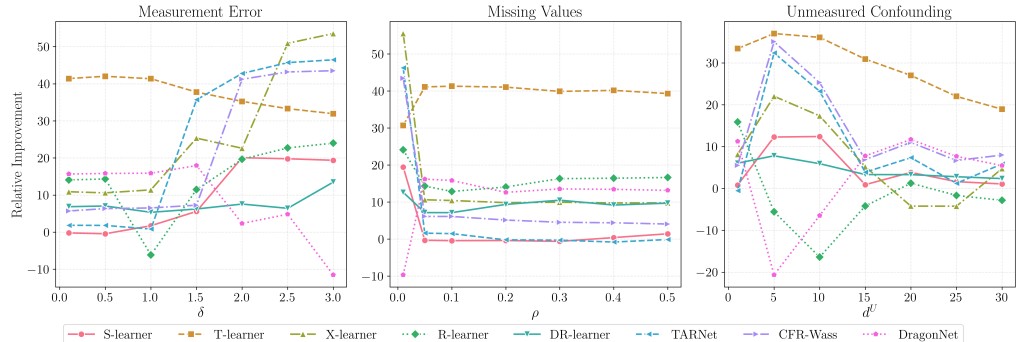

Figure 3: Performance gains from CARD: Relative Improvement in **worst-case PEHE** over 30 runs for various CATE learners under three distribution shift scenarios. The value is in percentage.

consistent gains under measurement error: relative improvements often exceed 10% for flexible learners such as TARNet and the R-learner at high noise levels, and TARNet's RI rises monotonically with noise to a peak of approximately 14% at $\delta = 2.5$. This pattern is intuitive: small amounts of noise leave a learner near its clean optimum so adversarial augmentation gives modest gains, whereas larger noise exposes vulnerabilities that the RL-guided diffusion discovers and the learner then learns to resist. By contrast, missing-value corruptions yield smaller but stable improvements, with RI ranging from 2%-9%, likely because the pipeline applies imputation (MICE) (Kallus et al., 2018), which already reduces extreme covariate variation and therefore narrows the space of harmful yet realistic augmentations. Unmeasured confounding is the most challenging regime: relative gains are smaller (0%-4%) but remain practically important because they help preserve performance when treatment heterogeneity itself shifts. Overall, these results show that CARD offers stronger defense against measurement error and missing values, and it also delivers consistent, constructive gains when unseen shifts arise from latent confounders.

**The magnitude of CARD's improvement is model-specific.** Results confirm that nearly every base learner benefits, and more interestingly, the magnitude of this improvement is heterogeneous. For instance, more flexible models like TARNet, R-learner, and X-learner are among the biggest beneficiaries, particularly under measurement error. This model-specific efficacy can be attributed to two main factors. First, learners possess different inductive biases (Curth & Van der Schaar, 2021). Models with more flexible function classes, such as TARNet, can better exploit the adversarial augmentations to learn more robust CATE functions. In contrast, simpler or heavily regularized learners may already exhibit some robustness, leaving less capacity for substantial improvement. Second, baseline vulnerability plays a key role. The T-learner, for example, which is highly susceptible to noise, still receives significant relative performance gains, suggesting that CARD effectively enhances resilience even when baseline errors are large.

### 5.2.2 Worst-Case Robustness Analysis

In addition to the analysis of average PEHE, we also investigate whether CARD can improve the worst-case PEHE, which is a critical measure of model stability and robustness. Relevant results are reported in Table 2 and Figure 3.

**CARD consistently enhances performance across diverse learners and shifts.** A key finding from Table 2 and Figure 3 is that CARD's impact on a model's worst-case performance is significantly larger than its effect on average performance, Averaged across learners, the mean worst-case RI is substantial for measurement-error scenarios (around 21.1%), moderate for missing-value scenarios (around 11.5%), and smaller but nontrivial for unmeasured confounding (around 8.0%). Notably, the RI in worst-case PEHE is frequently three to five times greater than the improvement observed in the average-case. For instance, while CARD consistently improves the T-learner's average PEHE by approximately 10% in many high-noise settings, it enhances its worst-case PEHE by a massive 30-40% under the same conditions. This disparity reveals CARD's primary mechanism: CARD not only shifts average behavior but also substantially reduces the tail risk.

**The magnitude of CARD's improvement is context-dependent.** Similar to average-case results, we also find CARD's capabilities are dependent on type of distribution shifts in worst-case. As shown in Figure 3, CARD delivers its most dramatic gains under measurement error, a scenario that often causes covariate shifts due to additive noise in standard models. Here, CARD slashes the worst-case PEHE of flexible learners CFR-Wass by 43.4% (from 9.834 to 5.565) and that of TARNet by 46.2% (from 8.715 to 4.686) at the highest noise level ($\delta = 3.0$). By contrast, missingness produces smaller but stable improvements, which is consistent with our previous observation. Under the more structured challenge of unmeasured confounding, the improvements, while smaller or sometimes negative, are still effective for enhancing the worst-case performance. For example, it reduces the T-learner's worst-case PEHE by a substantial 20%-30% at the highest confounding dimension. However, this results also highlights a fundamental problem in causal identification: while data augmentations can significantly improve a model's robustness, they cannot identify oracle causal information that is actually absent.

**The magnitude of CARD's improvement is model-specific.** The benefits of CARD are distinct across base learners. As illustrated in Figure 3, for instance, some learners exhibit large worst-case RI under measurement error: T-learner achieves worst-case RI averaged in all bias levels with about 36.0%, and X-learner with about 32.8%. Others show modest gains, such as DragonNet (5.1%) and S-learner (12.2%). This heterogeneity can be attributed to two complementary factors: (i) learners with higher worst-case baseline PEHE have more room for improvement; and (ii) flexible neural architectures like TARNet can leverage adversarial augmentations to learn more stable conditional effects. These two observations are aligned with previous average-case results. Interestingly, in a few cases (e.g., R-learner and DragonNet under certain hidden confounding levels), CARD produces marginally negative RI, which may be linked to their connections with targeted maximum likelihood estimation (TMLE), where CARD's perturbations interact with the targeted nuisance components.

## 6 Conclusion

In this work, we introduce CARD, a novel and model-agnostic framework that is capable to improve the robustness of any existing CATE learner to unknown distribution shift, without requiring prior knowledge or additional structural assumptions in the deployment domain. Rather than proposing a new CATE estimation algorithm, our primary goal is to investigate how reinforcement learning guided diffusion models can generate adversarial proxies that encourage the CATE learner to adapt and remain resilient to unseen distribution shifts. Experiments across diverse learners and distribution shift types show consistent gains from CARD, highlighting its potential effectiveness for real-world deployment. The limitation of this work lies in the computational complexity, a common challenge for diffusion models, as discussed in Section A.2. An interesting future research is the complexity improvement with recent acceleration techniques (Chen et al., 2024). Simultaneously, the success of this approach might open exciting future directions, including extending its application to other causal tasks related to generative modeling, such as counterfactual generation (Yoon et al., 2018), dimension reduction (Liu et al., 2024), and model evaluation (Athey et al., 2024).

**Reproducibility statement.** We defer the implementation details of using CARD to train CATE in Appendix A.1. The uploaded code can be directly used to reproduce our experimental results. Additionally, we list all the referred and required resources with an instruction file in supplementary.

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

## A APPENDIX

### A.1 CATE ESTIMATION WITH CARD

We now detail the construction of a CATE learner under the CARD framework, leveraging the observed samples $\{(X_i, A_i, Y_i)\}_{i=1}^n$. Since the CATE learner is trained on the training dataset, the sample size n here corresponds to the size of the training sample. We denote $n_t$ as the sample size of the treatment group and $n_c$ as that of the control group, with $n = n_t + n_c$. A key component of the CARD framework is a time-dependent diffusion model $g_\theta(z, t)$, which takes as inputs a time step $t \sim \mathcal{U}[\epsilon, T]$ and a noise variable $z \sim \mathcal{N}(0, 1)$. The diffusion model's reverse process initiates at time step $T$ and progresses iteratively. Upon reaching time step 0, it generates the latent variables $Z_0$.

- **S-learner**: Let the predictors be $(X, A)$ and the response be Y. We first initialize the model $\hat{\mu}(X, A)$ and then, under the CARD framework, employ $\|Y - \hat{\mu}(X \oplus Z_0, A)\|_2^2$ as both the loss function and reward function to co-optimize $\hat{\mu}(X, A)$ and the score-based diffusion model $g_\theta$ through an alternating training process. Using $Z_0$ generated by $g_\theta$, we obtain $\hat{\tau}_S(X)$:
$$\hat{\tau}_S(X) = \hat{\mu}(X \oplus Z_0, 1) - \hat{\mu}(X \oplus Z_0, 0).$$

- **T-learner**: Let the predictors be $X^T$ (covariates in the treatment) and the response be $Y^T$ (outcome in the treatment). Let the predictors be $X^C$ (covariates in the control) and the response be $Y^C$ (outcome in the control). We first initialize the treatment outcome model $\hat{\mu}_1(X^T)$ and control outcome model $\hat{\mu}_0(X^C)$. Under the CARD framework, we then employ $\|Y^T - \hat{\mu}_1(X^T \oplus Z_0)\|_2^2 + \|Y^C - \hat{\mu}_0(X^C \oplus Z_0)\|_2^2$ as both the loss function and reward function to co-optimize $\hat{\mu}_1$, $\hat{\mu}_0$, and the diffusion model $g_\theta$ through an alternating training process. Using $Z_0$ generated by $g_\theta$, we obtain $\hat{\tau}_T(X)$:
$$\hat{\tau}_T(X) = \hat{\mu}_1(X \oplus Z_0) - \hat{\mu}_0(X \oplus Z_0).$$

- **X-learner**: First-step: Initialize $\hat{\mu}_1(X)$ and $\hat{\mu}_0(X)$ using the the above-mentioned procedure in T-learner. Let the predictors be $X$ and the response be $A$. Initialize a propensity score model $\hat{\pi}(X)$. Second-step: Let the predictors be $X^T$ and the response be $\hat{\mu}_1(X^T) - Y^T$. Let the predictors be $X^C$ and the response be $\hat{\mu}_0(X^C) - Y^C$. Using these defined predictors and responses, we initialize the models $\hat{\tau}_1(X^T)$ and $\hat{\tau}_0(X^C)$. Next, we utilize $\left\| Y^T - \hat{\mu}_1(X^T \oplus Z_0) \right\|_2^2 + \left\| Y^C - \hat{\mu}_0(X^C \oplus Z_0) \right\|_2^2 + \text{CrossEntropyLoss}(X \oplus Z, A) + \left\| \hat{\mu}_1(X^T \oplus Z) - Y^T - \hat{\tau}_1(X^T \oplus Z_0) \right\|_2^2 + \left\| \hat{\mu}_0(X^C \oplus Z_0) - Y^C - \hat{\tau}_0(X^C \oplus Z_0) \right\|_2^2$ as both the loss function and reward function to co-optimize $\hat{\mu}_1$, $\hat{\mu}_0$, $\hat{\pi}$, $\hat{\tau}_1$, $\hat{\tau}_0$ and the diffusion model $g_\theta$ through an alternating training process. Using $Z_0$ generated by $g_\theta$, we obtain $\hat{\tau}_X(X)$:

$$\hat{\tau}_X(X) = (1 - \hat{\pi}(X \oplus Z_0))\hat{\tau}_1(X^T \oplus Z_0) - \hat{\pi}(X \oplus Z_0)\hat{\tau}_0(X^C \oplus Z_0).$$

- **R-learner**: First-step: Let the predictors be $X$ and the response be $Y$. Initialize a model $\hat{\mu}(X)$ to approximate the conditional mean outcome $\mathbb{E}[Y|X]$. Initialize a propensity score model $\hat{\pi}(X)$ using the the above-mentioned procedure in X-learner. Second-step: Compute the outcome residual $\xi = Y - \hat{\mu}(X)$ and treatment residual $\nu = T - \hat{\pi}(X)$. We then initialize a model $\hat{\tau}(X)$. Under the CARD framework, we utilize $\|Y - \hat{\mu}(X \oplus Z_0)\|_2^2 + \text{CrossEntropyLoss}(X \oplus Z_0, A) + \|\xi - \nu\hat{\tau}(X \oplus Z_0)\|_2^2$ as both the loss function and reward function to co-optimize $\hat{\mu}$, $\hat{\pi}$, $\hat{\tau}$ and the diffusion model $g_\theta$ through an alternating training process. Using $Z_0$ generated by $g_\theta$, we obtain $\hat{\tau}_R(X)$:

$$\hat{\tau}_R(X) = \hat{\tau}(X \oplus Z_0).$$

- **DR-learner**: First-step: Initialize $\hat{\mu}_1(X)$ and $\hat{\mu}_0(X)$ using the the above-mentioned procedure in T-learner. Initialize a propensity score model $\hat{\pi}(X)$ using the the above-mentioned procedure in X-learner. Second-step: Construct surrogate of CATE using pseudo-outcomes with doubly robust (DR) formula: $Y_{DR}^{0,1} = Y_{DR}^1 - Y_{DR}^0$, where $Y_{DR}^1 = \hat{\mu}_1(X) + \frac{T}{\hat{\pi}(X)}(Y - \hat{\mu}_1(X))$ and $Y_{DR}^0 = \hat{\mu}_0(X) + \frac{1-T}{1-\hat{\pi}(X)}(Y - \hat{\mu}_0(X))$. Using these defined predictors and responses, we initialize the models $\hat{\tau}(X)$. Next, we utilize $\left\| Y^T - \hat{\mu}_1(X^T \oplus Z_0) \right\|_2^2 + \left\| Y^C - \hat{\mu}_0(X^C \oplus Z_0) \right\|_2^2 + \text{CrossEntropyLoss}(X \oplus Z_0, A) + \left\| Y_{DR}^{0,1} - \hat{\tau}(X \oplus Z_0) \right\|_2^2$ as both the loss function and reward function to co-optimize $\hat{\mu}_1$, $\hat{\mu}_0$, $\hat{\pi}$, $\hat{\tau}$ and the diffusion model $g_\theta$ through an alternating training process. Using $Z_0$ generated by $g_\theta$, we obtain $\hat{\tau}_{DR}(X)$:

$$\hat{\tau}_{DR}(X) = \hat{\tau}(X \oplus Z_0).$$

- **TARNet**: We first define the predictors as $(X, A)$ and the response as Y, and construct a representation model $\hat{r}(X)$ to encode covariate information. The model architecture incorporates two outcome heads: $\hat{\mu}_1(\hat{r}(X))$ for the treatment group and $\hat{\mu}_0(\hat{r}(X))$ for the control group, which share the underlying representation $\hat{r}(X)$ while learning separate outcome estimates. Under the CARD framework, we employ the composite function $\left\| \hat{\mu}_1(\hat{r}(X \oplus Z_0)) - Y^T \right\|_2^2 + \left\| \hat{\mu}_0(\hat{r}(X \oplus Z_0)) - Y^C \right\|_2^2$ as both the loss function and reward function to co-optimize $\hat{r}(X)$, $\hat{\mu}_1$, $\hat{\mu}_0$, and the diffusion model $g_\theta$ through an alternating training process. Using $Z_0$ generated by $g_\theta$, we obtain $\hat{\tau}_{TARNet}(X)$:

$$\hat{\tau}_{TARNet}(X) = \hat{\mu}_1(\hat{r}(X \oplus Z_0)) - \hat{\mu}_0(\hat{r}(X \oplus Z)).$$

- **CFR_WASS**: Initialize $\hat{r}(X)$, $\hat{\mu}_1(X)$ and $\hat{\mu}_0(X)$ using the the above-mentioned procedure in TARNet. Under the CARD framework, we employ the composite function $\left\| \hat{\mu}_1(\hat{r}(X \oplus Z_0)) - Y^T \right\|_2^2 + \left\| \hat{\mu}_0(\hat{r}(X \oplus Z_0)) - Y^C \right\|_2^2 + \text{IPMLoss}(X^T, X^C)$ as both the loss function and reward function to co-optimize $\hat{r}(X)$, $\hat{\mu}_1$, $\hat{\mu}_0$, and the diffusion model $g_\theta$ through an alternating training process. Using $Z_0$ generated by $g_\theta$, we obtain $\hat{\tau}_{CFR_W ASS}(X)$:

$$\hat{\tau}_{CFR-Wass}(X) = \hat{\mu}_1(\hat{r}(X \oplus Z_0)) - \hat{\mu}_0(\hat{r}(X \oplus Z_0)).$$

- **DragonNet**: Initialize $\hat{r}(X)$, $\hat{\mu}_1(X)$ and $\hat{\mu}_0(X)$ using the the above-mentioned procedure in TARNet. The model architecture incorporates three outcome heads: $\hat{\mu}_1(\hat{r}(X))$ for the

treatment group, $\hat{\mu}_0(\hat{r}(X))$ for the control group and $\hat{\pi}(X)$ for the propensity score, which share the underlying representation $\hat{r}(X)$ while learning separate outcome estimates. Under the CARD framework, we employ the composite function $\left\|\hat{\mu}_1(\hat{r}(X \oplus Z_0)) - Y^T\right\|_2^2 + \left\|\hat{\mu}_0(\hat{r}(X \oplus Z_0)) - Y^C\right\|_2^2 + \text{CrossEntropyLoss}(X \oplus Z_0, A)$ as both the loss function and reward function to co-optimize $\hat{r}(X)$, $\hat{\mu}_1$, $\hat{\mu}_0$, and the diffusion model $g_\theta$ through an alternating training process. Using $Z_0$ generated by $g_\theta$, we obtain $\hat{\tau}_{DragonNet}(X)$:

$$\hat{\tau}_{DragonNet}(X) = \hat{\mu}_1(\hat{r}(X \oplus Z_0)) - \hat{\mu}_0(\hat{r}(X \oplus Z_0)).$$

## A.2  EXPERIMENTAL DETAILS AND HYPERPARAMETERS

**Implementation details.**  All meta-learners in this work are implemented using neural network architectures. Specifically, the S-learner, T-learner, X-learner, R-learner, and DR-learner share a unified three-layer neural network structure, with each layer containing 200 neurons. In contrast, TARNet, CFR-Wass, and DragonNet adopt a two-component architecture: a representation network with three layers (200 neurons per layer) and a prediction layer with three layers (100 neurons per layer).

**Hyperparameters.**  All model training processes are conducted on a Dell 3640 workstation with an Intel Xeon W-1290P 3.60GHz CPU and NVIDIA GeForce RTX 2080 Ti GPU. For optimizing the CATE learner, we used the Adam optimizer with a learning rate of $10^{-3}$ and weight decay of $10^{-4}$. Model selection was based on the factual loss as the validation metric, with early stopping implemented if no improvement was observed on the validation set for 20 consecutive epochs. In the reinforcement fine-tuning phase, the AdamW optimizer was employed with a learning rate of $2 \times 10^{-5}$. The hyperparameters for fine-tuning varied by model type:

- For S-learner, T-learner, and DragonNet: $\alpha = 0.8$ and fine-tuning frequency $K = 10$;
- For X-learner, R-learner, and DR-learner: $\alpha = 0.8$ and $K = 2$;
- For TARNet and CFR_WASS: $\alpha = 0.1$ and $0.8$, with $K = 10$ and $5$ respectively.

Additionally, the imbalance loss coefficient for CFR-Wass and the BCE loss coefficient for DragonNet were both set to $1.0$. The discount factor $\gamma$ was set to $0.99$. The latent variable $Z$ generated by the diffusion model has a dimension half that of the covariate $X$.

**Model architecture.**  The parameters of the autoencoder and score-based diffusion model largely follow the default settings provided in (Suh et al.). Both models are trained for 10,000 epochs, and the number of timesteps for the diffusion model is set to 50. The autoencoder adopts a multi-layer perceptron (MLP) block-based architecture, with ReLU activation functions used in all hidden layers. Its forward process is defined as:

$$\begin{aligned}
\text{MLPBlock}(X) &= \text{ReLU}\left(\text{Linear}(X)\right), \\
Z &= \text{Linear}\left(\cdots \text{MLPBlock}(X)\right), \\
\tilde{X} &= \text{Linear}\left(\cdots \text{MLPBlock}(Z)\right),
\end{aligned} \tag{9}$$

where $Z$ denotes the latent representation of the input $X$, and $\tilde{X}$ is the reconstruction output of the autoencoder.

Let $t$ denote a timestep in the diffusion process, and SinTimeEmb represent the sinusoidal time embedding proposed in (Nichol & Dhariwal, 2021). For any fixed $t$, the time embedding $t^{\text{emb}}$ and the processed input to the score network (denoted $Z^{t\text{-emb}}$) are computed as:

$$\begin{aligned}
t^{\text{emb}} &= \text{LayerNorm}\left(\text{SiLU}\left(\text{Linear}\left(\text{SinTimeEmb}(t)\right)\right)\right), \\
Z^{t\text{-emb}} &= \text{LayerNorm}\left(\text{Linear}(Z_t)\right) + t^{\text{emb}},
\end{aligned} \tag{10}$$

where $Z_t$ is the latent variable at timestep $t$, and the addition of $t^{\text{emb}}$ injects timestep-aware information into the latent input.

The time-dependent score network $g$ is then constructed using MLP blocks with LayerNorm regularization, and its calculation is given by:

$$\text{MLPBlock}(Z^{t\text{-emb}}) = \text{LayerNorm}\left(\text{ReLU}\left(\text{Linear}(Z^{t\text{-emb}})\right)\right),$$
$$g(Z^{t\text{-emb}}, t) = \text{Linear}\left(\cdots \text{MLPBlock}(Z^{t\text{-emb}})\right). \tag{11}$$

**Time complexity analysis.** We compare the time complexity between CATE learners trained with the CARD framework and those trained with standard procedures. We assume the CATE learner is trained with E epochs, resulting in a complexity of $O(E)$. In contrast, when training a CATE learner using the CARD framework, each training epoch requires an additional T iterations for trajectory generation in the diffusion model, leading to a time complexity of $O(ET)$. Thus, the improved robustness of the CATE learner achieved via the CARD framework comes at the cost of increased computational time, i.e., a tradeoff between model robustness and time cost. The practical users are suggested to set $T$ with early stop, and use new acceleration technique for training diffusion models.

### A.3 ROLE OF LLM

In this paper, LLM was used to aid in writing and polish the texts. Importantly, we take full responsibility for the content of the manuscript, and we did not use LLM for idea generation, method development, experimental coding. All research ideas, codes, experimental results, and experimental analysis are conducted by the authors. The contribution of LLM is only the linguistic quality improvement.

