# OpenReview forum: "Improving Causal Inference Robustness via Reinforcement-guided Diffusion Models"
_ICLR.cc/2026/Conference — ICLR 2026 Conference Withdrawn Submission_

### Official Review · Reviewer_2Bun · 2025-10-20

**Soundness:** 2
**Presentation:** 3
**Contribution:** 2
**Rating:** 4
**Confidence:** 4

**Summary:**

This paper introduces a "model-agnostic" framework named Causal Adversarial Reinforcement-guided Diffusion (CARD), designed to address the performance degradation of CATE (Conditional Average Treatment Effect) estimation models when facing unknown distributional shifts between the training and deployment environments. CARD formulates CATE modeling as a minimax game: a reinforcement learning (RL) agent guides a diffusion model to generate adversarial data augmentations to maximize the CATE learner's loss; subsequently, the CATE learner is trained by minimizing this worst-case loss, thereby achieving robust optimization.

**Strengths:**

- The authors propose a "model-agnostic" framework named CARD, which can be flexibly wrapped around any existing CATE learner without redesigning its internal architecture.

- This paper is the first to introduce Reinforcement Learning (RL)-guided diffusion models into the causal inference literature, providing a new perspective for achieving robust optimization of CATE estimation.

**Weaknesses:**

- The measurement error (Gaussian noise) and missing value simulations in this paper's evaluation, whose mechanisms align with CARD's adversarial training, might inherently favor CARD. However, in the more severe "unmeasured confounding" tests, CARD's benefits significantly diminish. This suggests that the experimental design may have exaggerated its robustness and failed to reveal its limitations in addressing changes to the true causal structure.

- The authors' contribution lies in the robust optimization algorithm, rather than solving the CATE "identifiability" challenge. The authors acknowledge that data augmentation cannot identify missing prior causal information. Therefore, when faced with severe unmeasured confounding, CARD can enhance stability, but its ability to correct fundamental bias is limited by this theoretical boundary.

- The proposed framework has a significant drawback: a lack of clear validity boundaries. Its flexible, learning-based adversarial agent, while avoiding predefined uncertainty sets, also leads to insufficient theoretical characterization. The manuscript fails to provide formal guarantees or sensitivity analysis to define the conditions under which CARD ensures performance improvement.

- High computational cost affects deployment, which the authors acknowledge as a major limitation. CARD's robustness comes at the expense of computation time. Training complexity increases from $O(E)$ to $O(ET)$due to the diffusion model, posing a barrier for large-scale data or real-time applications.

- In Table 2, the authors state that bolding marks "better results" (lower PEHE), but this rule is not followed. In key comparisons, such as the S-learner, the incorrectly bolded "+CARD" variants actually have higher PEHE (worse performance). This confusing presentation reduces readability and misleads the reader.

**Questions:**

See Weaknesses

---

### Official Review · Reviewer_3gKX · 2025-10-29

**Soundness:** 2
**Presentation:** 2
**Contribution:** 2
**Rating:** 2
**Confidence:** 4

**Summary:**

This paper proposes CARD, a model-agnostic framework using an adversarial, reinforcement-guided diffusion model with the goal of making CATE estimation more robust in a setting with source data where the CATE can be identified and estimated, and target data where estimation of the CATE is challenging or identification might not be possible due to measurement error, missing values, and unobserved confounding. The authors provide results of their method under different data-corruption settings.

**Strengths:**

- Robust CATE estimation under changes between training and test environments is an important topic.
- Most of the paper is well motivated, and the idea of the method is easy to follow. In general, the paper is well structured and nicely written.
- The authors provide code for reproducibility.

**Weaknesses:**

- **Problem setting and “concept drift” remain unclear.** The main focus appears to be on violations of transportability (usually effect modifiers. However, in the motivating example, it sounds more like certain confounders fully observed in the source dataset are not observed in the target dataset; however, the described setting implies that the same covariates \(X\) are observed in both datasets. Thus, the violation of transportability seems more likely to arise due to effect modifiers (as described in prior work), for example from different treatment assignments in both domains (with the simplest case being RCTs in the source domain to ensure identifiability here). The setting and motivation should be formulated more clearly. In Sec. 3, the authors should also summarize and formalize at the end which exact setup (data, assumptions, and violations of assumptions) they consider for their application, including challenges mentioned in the introduction besides unobserved confounding (such as measurement error and missing values).
- **Lack of theoretical guarantees for unobserved confounding.** There are no guarantees or arguments for why the approach helps with unobserved confounding (no sensitivity model), no partial identifiability results, and little intuition for when improvements should be expected. Without a specific sensitivity-model formulation, this seems difficult to show; thus the work appears primarily heuristic and demonstrated only in limited settings.
- **Contribution appears limited.** Since the causal challenge of unobserved confounding is not directly tackled by the method (beyond using CATE estimators), the contribution seems limited and appears to be a straightforward extension of prediction tasks.
- **Insufficient experimental setup.** Missing values do not introduce any distribution shift when values are missing completely at random. Also, to introduce unobserved confounding, it is not clear from the potential-outcomes function in Eq. 8 how strongly this affects the treatment effect (it could average out in expectation or even cancel out).
- **Empirical results are counterintuitive.** For example, in Figure 2, why should CARD lead to performance gains without any missing values or unobserved confounding, and then the gains decrease as more missing values/unobserved confounding are introduced? This seems at odds with the paper’s motivation.
- **Related work should be extended.** Since this paper uses unobserved confounding as a main part of their motivation, important work around how to tackle unobserved confounding, especially around sensitivity analysis and partial identification (see e.g. https://proceedings.neurips.cc/paper_files/paper/2023/file/7f8b8bc8ebac661c442c4dafd5d98c08-Paper-Conference.pdf and mentioned related work there, or also the instrument setting) should be added. This is particularly important to understand the main problem of this paper that point identification is not achievable in such a setting (without any strict additional assumptions).

## Overall
Trying to show improvements in point identification under potential unobserved confounding seems like a flawed setup. Providing more experiments alone is not sufficient to demonstrate the practicality of this work.

**Questions:**

- How can the counterintuitive patterns in the results, such as those in Figure 2, be explained?
- Are there any theoretical guarantees or intuitions for how CARD can improve performance in the considered settings, especially under unobserved confounding?

---

### Official Review · Reviewer_4L6u · 2025-11-01

**Soundness:** 3
**Presentation:** 3
**Contribution:** 3
**Rating:** 6
**Confidence:** 2

**Summary:**

The paper Improving Causal Inference Robustness via Reinforcement-Guided Diffusion Models presents Causal Adversarial Reinforcement-guided Diffusion (CARD), a model-agnostic framework that improves the robustness of Conditional Average Treatment Effect (CATE) estimators under unknown distribution shifts. CARD treats robust causal estimation as a minimax optimization problem, where a reinforcement learning agent guides a diffusion model to generate adversarial data augmentations that expose weaknesses in the CATE learner, prompting it to adapt and minimize the worst-case error. This approach strengthens robustness without relying on extra structural assumptions or target-domain information. The work also introduces the use of reinforcement-guided diffusion models to causal inference, opening possibilities for future applications such as counterfactual generation and model evaluation.

**Strengths:**

1. Proposes a general, model-agnostic framework (CARD) that can be integrated with any existing CATE learner without redesigning its architecture.

2. Introduces the novel use of reinforcement-guided diffusion models in causal inference, providing a new direction for robust estimation.

3. Formulates CATE robustness as a principled minimax optimization problem, offering a clear theoretical foundation for adversarial training.

**Weaknesses:**

1. The minimax optimization framework introduces considerable computational cost, mainly because of the diffusion model’s iterative sampling process.

2. The reinforcement-guided adversarial generation lacks formal theoretical guarantees for convergence and stability within the minimax formulation.

3. The method assumes that CATE learners are differentiable and smooth, which restricts its use with non-differentiable or discrete models.

4. The framework’s performance depends on the quality of the latent representations learned by the autoencoder, which may introduce bias if the encoding is inadequate.

**Questions:**

1.  How stable is the joint training process between the diffusion model and the CATE learner?

2.  Since the diffusion model introduces heavy computation, can lighter generative alternatives achieve similar effects with lower cost?

3.  The method assumes access to a well-trained autoencoder for latent representation—could integrating end-to-end training or representation regularization improve stability?

4.  How sensitive is CARD to the reward design in the reinforcement learning component, and would a simpler heuristic adversarial signal suffice?

---

### Official Review · Reviewer_mC8T · 2025-11-02

**Soundness:** 1
**Presentation:** 1
**Contribution:** 1
**Rating:** 2
**Confidence:** 4

**Summary:**

The work proposes a new method to ensure robustness to covariate shifts for conditional average treatment effect (CATE) estimation, namely, Causal Adversarial Reinforcement-guided Diffusion (CARD). The method is based on a minimax game between an arbitrary CATE learner (which aims to learn CATE in a distributionally robust way) and a diffusion model (that adversarially generates augmentations to the covariates). In this way, the CARD implements a distributionally robust CATE estimation. To demonstrate the effectiveness of the CARD, the authors have evaluated it on several semi-synthetic benchmarks with covariate shifts (e.g., measurement errors, missing values, and unmeasured confounding).

**Strengths:**

The paper studies a relatively underexplored problem of causal machine learning: robustness to covariate shifts and external validity of CATE learners. The proposed method is somewhat original.

**Weaknesses:**

However, I found multiple fundamental flaws in the current version of the paper:

- **Lack of theoretical guarantees**. The suggested method promises to address multiple issues of the covariate shift (i.e., (a) measurement errors, (b) missing values, and (c) unmeasured confounding). Yet, **no theoretical assumptions** were provided on (1) how the  CARD tackles all these settings, and (2) whether CATE is identifiable at all (with any method), given target domain data. For example, the authors have used the approach of [1] for setting (b) missing values, but provided no discussion on the underlying assumptions of [1]. Although some assumptions were offered in Sec. 3.1, they do not directly relate to (a)-(c) but rather to the absence/presence of the effect modifiers between source/target domains. Therefore, I fail to see how the proposed method achieves consistent estimation in all the settings (a)-(c). I understand that the experimental evidence suggests a high effectiveness of the method, yet the authors cannot claim that the method works for any general type of (a) measurement errors, (b) missing values, and (c) unmeasured confounding. For example, given unmeasured confounding in the target domain data, the CATE is not point-identifiable, and I fail to see how CARD solves this issue.

- **Unnecessary complexity of the method**. In my opinion, the CARD method is unnecessarily complex. Why can’t we just use a diffusion (or any other generative) model without the whole RL machinery to adversarially generate $Z$? I suggest the authors provide such a justification. Also, ablation studies would be appreciated.

- **Missing baselines**. I think mentioned in the paper baseline of [2] can be easily adapted to the CATE estimation setting, and [3] is specifically designed to enforce the distributional robustness for CATE estimation. I encourage the authors to include [2, 3] as baselines for all the benchmarks.

Apart from the major issues listed above, I found small mistakes in the paper:

- Not all the notation is clearly defined. What is $\bigoplus$ and how is $\Omega$ defined?
- Line 250. There is no Enc in Eq. (5).
- Line 208. Missing date in (Black et al.) paper.

References:
- [1] Nathan Kallus, Xiaojie Mao, and Madeleine Udell. Causal inference with noisy and missing covariates via matrix factorization. Advances in neural information processing systems, 31, 2018.
- [2] Nathan Kallus, Xiaojie Mao, Kaiwen Wang, and Zhengyuan Zhou. Doubly robust distributionally robust off-policy evaluation and learning. In International Conference on Machine Learning, pp. 10598–10632. PMLR, 2022.
- [3] Yiyan Huang, Cheuk Hang Leung, Siyi Wang, Yijun Li, and Qi Wu. Unveiling the potential of robustness in evaluating causal inference models. In Advances in Neural Information Processing Systems, 2024.

**Questions:**

1. I didn’t actually understand: Which assumption is assumed in Sec. 3.1: covariate shift or concept drift?
2. I wonder whether the works on generalization bounds [1, 2] for CATE estimation can be adapted to the setting of the paper? For example, we can define an adversarial perturbation in terms of the counterfactual variance [1] or chi-squared distance [2]. Should they also be considered as baselines?

References:
- [1] Yao Zhang, Alexis Bellot, and Mihaela van der Schaar. Learning overlapping representations for the estimation of individualized treatment effects. In International Conference on Artificial Intelligence and Statistics, 2020.
- [2] Daniel Csillag, Claudio Jose Struchiner, and Guilherme Tegoni Goedert. Generalization bounds for causal regression: Insights, guarantees and sensitivity analysis. In International Conference on Machine Learning, 2024.

---

### Note · Authors · 2026-01-21

I have read and agree with the venue's withdrawal policy on behalf of myself and my co-authors.